# Speaking Your Language: Spatial Relationships in Interpretable Emergent Communication

**Olaf Lipinski**[1][*]     **Adam J. Sobey**[2,1]     **Federico Cerutti**[3]     **Timothy J. Norman**[1]
[1]University of Southampton     [2]The Alan Turing Institute     [3]University of Brescia
{o.lipinski,t.j.norman}@soton.ac.uk
asobey@turing.ac.uk
federico.cerutti@unibs.it

## Abstract

Effective communication requires the ability to refer to specific parts of an observation in relation to others. While emergent communication literature shows success in developing various language properties, no research has shown the emergence of such positional references. This paper demonstrates how agents can communicate about spatial relationships within their observations. The results indicate that agents can develop a language capable of expressing the relationships between parts of their observation, achieving over 90% accuracy when trained in a referential game which requires such communication. Using a collocation measure, we demonstrate how the agents create such references. This analysis suggests that agents use a mixture of non-compositional and compositional messages to convey spatial relationships. We also show that the emergent language is interpretable by humans. The translation accuracy is tested by communicating with the receiver agent, where the receiver achieves over 78% accuracy using parts of this lexicon, confirming that the interpretation of the emergent language was successful.

## 1   Spatial referencing in emergent communication

Emergent communication allows agents to develop bespoke languages for their environment. While there are many successful examples of efficient (Rita et al., 2020) and compositional (Chaabouni et al., 2020) languages, they often lack fundamental aspects seen in human language, such as syntax (Lazaridou and Baroni, 2020) or recursion (Baroni, 2020). It is argued that these aspects of communication are important to improve the efficiency and generalisability of emergent languages (Baroni, 2020; Boldt and Mortensen, 2024; Rita et al., 2024). However, the current architectures, environments, and reward schemes are yet to exhibit such fundamental properties.

One such aspect is the development of *deixis* (Rita et al., 2024), which has been described as a way of pointing through language. Examples of *temporal deixis* include words such as "yesterday" or "before," and *spatial deixis* include words such as "here" or "next to" (Lyons, 1977). In emergent communication, Lipinski et al. (2023) investigate how agents may refer to repeating observations, which could also be viewed from the linguistic perspective as investigating *temporal deixis*. However, while there are advocates to investigate how emergent languages can develop key concepts from human language (Rita et al., 2024), no work has demonstrated the emergence of relative references to specific locations *within* an observation, or *spatial deixis*.

Spatial references would be valuable in establishing shared context between agents, increasing communication efficiency by reducing the need for detailed descriptions, and adaptability, by removing the need for unique references per object. For example, instead of describing a new, previously

---

[*]Corresponding author:  o.lipinski@soton.ac.uk

38th Conference on Neural Information Processing Systems (NeurIPS 2024).

unseen object, such as "a blue vase with intricate motifs on the table," one could simply use spatial relationships and say "the object left of the plate." Spatial referencing streamlines communication by leveraging the shared environment as a reference point. In dynamic environments where objects might change positions, spatial references enable agents to easily track and refer to objects without having to update their descriptions. This enhances communication efficiency and improves interaction and collaboration between agents. These elements may also help the evolved language become human interpretable, allowing the development of trustworthy emergent communication (Lazaridou and Baroni, 2020; Mu and Goodman, 2021).

This paper therefore explores how agents can develop communication with spatial references. While Rita et al. (2024) posit that the emergence of these references might require complex settings, we show that even agents trained in a modified version of the simple referential game (Lazaridou et al., 2018; Lewis, 1969) can develop spatial references.[2] This resulting language is segmented and analysed using a collocation measure, Normalised Pointwise Mutual Information (NPMI) adapted from computational linguistics. NPMI allows us to measure the strength of associations between message parts and their context, making it a valuable tool for gaining insights into the underlying structure of the emergent language. Using NPMI, we show how the agents compose such spatial references, providing the first hint of a syntactic structure, and showing that the emergent language can be interpreted by humans.

## 2 Development of a spatial referential game

Current emergent communication environments have not produced languages incorporating spatial references. To address this, we present a referential game (Lazaridou et al., 2018) environment where an effective language requires communication about spatial relationships.

### 2.1 Referential game environment

In the referential game, there are two agents, a sender and a receiver. The sender observes a vector and transmits its compressed representation through a discrete channel to the receiver. The receiver observes a set of vectors and the sender's message. One of these vectors is the same as the one the sender has observed. The receiver's goal is to correctly identify the vector the sender has described, among other vectors referred to as distractors. The simplicity of the referential games enables the reduction of extraneous factors which could impact the emergence of spatial references, such as transfer learning of the vision network or exploring action spaces in more complex environments.

In this work, the sender's input is an observation in the form of a vector $o = [o_1, o_2, o_3, o_4, o_5]$, where $\forall o \in \{-1, 0, 1 \ldots 59\}$. The vector $o$ is always composed of 5 integers. The observation includes a $-1$ in only one position, *e.g.*, $o_3 = -1$ for $o = [x, x, -1, x, x]$, to indicate the target integer for the receiver to identify. $o$ represents a window into a longer sequence $s$, which is randomly generated using the integers $\{0 \ldots 59\}$ without repetitions. This sequence is visible to the receiver, but **not** to the sender. As the target's position in the sequence is unknown to the sender, it has to rely on the relative positional information present in its observation, necessitating the use of *spatial referencing*.

Due to the window into the sequence being of length 5, it is necessary to shift the window when it approaches either extent of the sequence. The window is then shifted to the other side, maintaining the size of 5. For example, given a short sequence $s = [7, 5, 2, 12, 10, 4, 3, 15, 16, 13, 14, 6, 9, 8, 11, 1]$, if the selected target is 1, since there are no integers to the right of 1 the vector $o$ would be $o = [6, 9, 8, 11, -1]$ where it is shifted to the left as it approaches this rightmost extent of the sequence.

Due to the necessity of maintaining the window size, some observations provide additional positional information to the sender agent. Given the same example sequence $s$, we can categorise all observations into 5 types. The *begin* and *begin+1*, where the target integer is either at, or one after, the beginning of the sequence, *i.e.*, $o = [-1, 5, 2, 12, 10]$ or $o = [7, -1, 2, 12, 10]$. The *end* and *end-1*, where the target integer is either at, or one before, the end of the sequence, *i.e.*, $o = [6, 9, 8, 11, -1]$ or $o = [6, 9, 8, -1, 1]$. The most common case is the *middle* observation, where the target integer is anywhere in the sequence, excluding the first, second, second to last, and last positions, *e.g.*, $o = [12, 10, -1, 3, 15]$. Given a window of length 5, only 4 specific target integer positions per sequence can result in the other observations (*begin*, *begin+1*, *end-1*, and *end*). All other target

---

[2]Our code is available on GitHub at https://github.com/olipinski/TPG

integer positions within the sequence fall into the *middle* category, as they do not occupy the first, second, second to last, or last positions. Consequently, the majority of the target integer positions result in a *middle* type observation.

The sender's output is a message defined as a vector $\boldsymbol{m} = [m_1, m_2, m_3]$, where $m \in \{1 \dots 26\}$. 26 is chosen to allow for a high degree of expressivity, with the agents being able to use over 17k different messages, while also matching the size of the Latin alphabet. Since such a vocabulary size is enough to convey any information in natural languages like English, we consider that this should also apply to the agents. The vector $\boldsymbol{m}$ is always composed of 3 integers.

The receiver's input is an observation consisting of three vectors: the sender's message $\boldsymbol{m}$, the sequence $\boldsymbol{s}$, and the set of distractor integers together with the target integer $\boldsymbol{td}$. The distractor integers are randomly generated, without repetitions, given the same range of integers as the original sequence $\boldsymbol{s}$, *i.e.*, $\{0 \dots 59\}$, excluding the target object itself. Given an environment with 3 distractors, $\boldsymbol{td}$ could be $[d_1, t, d_2, d_3]$, where $t$ is the target object and $d_1, d_2, d_3$ are distractor objects. The position of the target object in $\boldsymbol{td}$ is randomised.

For example, given the sequence $\boldsymbol{s} = [7, 5, 2, 12, 10, 4, 3, 15, 16, 13, 14, 6, 9, 8, 11, 1]$, and the sender's observation $\boldsymbol{o} = [4, 3, -1, 16, 13]$, the vector $\boldsymbol{td}$ could be $\boldsymbol{td} = [7, 15, 11, 9]$, with 15 being the target that the receiver needs to identify. The sender could produce a message $\boldsymbol{m} = [3, 1, 1]$, which would mean that the target integer is one after the integer 3. This message would then be passed to the receiver, together with $\boldsymbol{s}$ and $\boldsymbol{td}$. The receiver would then have to correctly understand the message $\boldsymbol{m}$ (*i.e.*, that the target is one after 3) and find the integer 3 together with the following integer in the sequence $\boldsymbol{s}$. Having identified the target 15 given the message $\boldsymbol{m}$ and the sequence $\boldsymbol{s}$, it would output the correct position of this target in the $\boldsymbol{td}$ vector, *i.e.*, 2, since $\boldsymbol{td}_2 = 15$.

## 2.2 Spatial reference formalisation

To provide a generalisation of our results, we formalise what we refer to as spatio-temporal references. Let $O$ represent an abstract observation that an agent perceives from its environment, $O \in \mathbb{R}^m$, where $m$ represents the dimensions of the observation. For a 3D observation, $m$ could be $m = j \times k \times d$. Such an $m$ could represent a $j \times k$ matrix of $d = 3$ values, which, for example, could be an RGB picture, with $j \times k$ pixels and one value for each of the RGB colours ($d = 3$). The $m$ dimensions can represent the spatial, temporal, or other positions.

Let $O_p$ and $O_t$ be the coordinates of some elements in $O$, represented by an $m$-tuple of natural numbers $(x_1, x_2 ... x_m)$ and $(y_1, y_2 ... y_m)$, respectively. $O_p$ represents the reference point and $O_t$ represents a target point.

Then, the relative distance function $d(O_p, O_t)$ returns an $m$-tuple of integers $(z_1, z_2 ... z_m)$, such that $z_i = x_i - y_i$. This relative distance function allows for unambiguous identification of the target object $O_t$, given that the position of $O_p$ is known.

We define the spatio-temporally referent expression as a mapping of the value of $d(O_p, O_t)$, the reference point $O_p$, and their context $O$, to a specific linguistic or symbolic phrase that describes the relationship between $O_p$ and $O_t$. This mapping can be represented as:

$$(O, d(O_p, O_t), O_p) \rightarrow \text{Phrase}(O, d(O_p, O_t), O_p)$$

where the resulting expression $\text{Phrase}(O, d(O_p, O_t), O_p)$ is a description of the reference point $O_p$ and its relative distances to the target point $O_t$, given the context $O$.

The version of spatial referencing in our environment is a specific case of the general spatial reference formalisation, where the observation $O$ is represented as a one-dimensional tensor, and the target point $O_t$ is always indicated by the value $-1$ within the tensor. The sender's task is to describe the relative position of the target $O_t$ within this sequence, using a message that effectively communicates the spatial relationship between a chosen $O_p$ and the target $O_t$.

## 3 Agent Architecture

The agent architecture follows that of the most commonly used EGG agents (Kharitonov et al., 2019). This architecture is used to maintain consistency with the common approaches in emergent communication research (Chaabouni et al., 2019, 2020; Kharitonov et al., 2019; Lipinski et al., 2023;

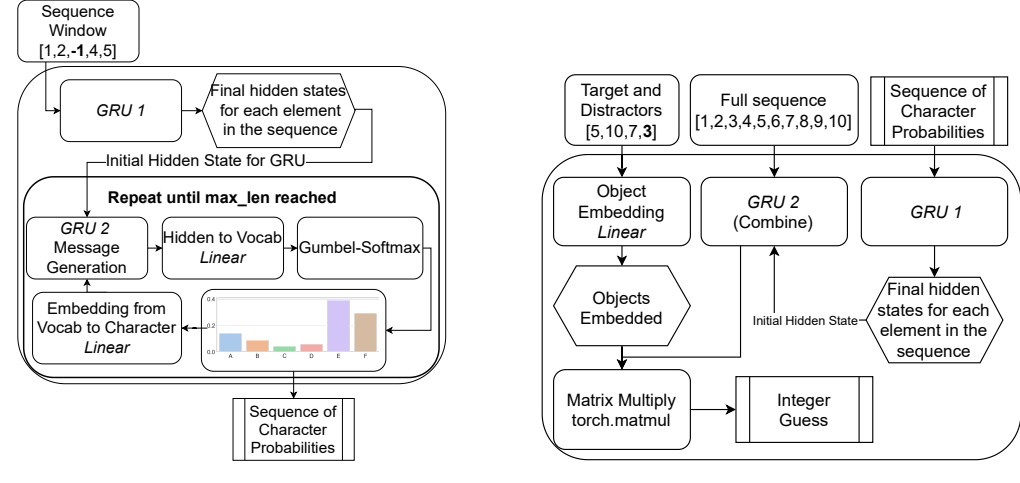

(a) The sender architecture.

(b) The receiver architecture.

Figure 1: The sender and receiver architectures. Adapted from (Lipinski et al., 2023).

Ueda and Washio, 2021), increasing the generalization of the results presented in this work. All environmental observations, *i.e.*, $o$, $s$, and $td$, are passed in as scalars, as one-hot encoding of the observation vectors leads to agents memorising the dataset.

The sender agent, shown in Figure 1a, receives a single input, the vector $o$, which is passed through the first GRU of the sender. The resulting hidden state is used as the initial hidden state for the message generation GRU (Cho et al., 2014). The message generation GRU is used to produce the message, character by character, using the Gumbel-Softmax reparametrization trick (Jang et al., 2017; Kharitonov et al., 2019; Mordatch and Abbeel, 2018). The sequence of character probabilities generated from the sender is used to output the message $m$.

$m$ is input to the receiver agent, shown in Figure 1b, together with the full sequence $s$ and the target and distractors $td$. The message is processed by the first receiver GRU, which produces a hidden state used as the initial hidden state for the GRU processing the sequence $s$. This is the only change from the standard EGG architecture (Kharitonov et al., 2019). This additional GRU allows the receiver agent to process the additional input sequence $s$, using the information contained within the message $m$. The goal of this GRU is to use the information provided by the sender to correctly identify which integer from the sequence $s$ is the target integer. The final hidden state from the additional GRU is multiplied with an embedding of the targets and distractors, to output the receiver's prediction. This prediction is in the form of the index of the target within $td$.

Following the commonly used approach (Kharitonov et al., 2019), agent optimisation is performed using the Gumbel-Softmax reparametrization (Jang et al., 2017; Mordatch and Abbeel, 2018), allowing for direct gradient flow through the discrete channel. The agents' loss is computed by applying the cross entropy loss, using the receiver target prediction and the true target label. The resulting gradients are passed to the Adam optimiser and backpropagated through the network. Detailed training hyperparameters are provided in Appendix A.

## 4   Message interpretability and analysis using NPMI

To analyse spatial references in emergent language, a way to identify their presence is essential. In discrete emergent languages, interpretation is typically done by either using dataset labels in natural language (Dessì et al., 2021), or by qualitative analysis of specific messages (Havrylov and Titov, 2017). However, both of these techniques require message-meaning pairs, and so neither would be able to identify the presence of spatial references, as the labels for spatial relationships that the agents refer to would not necessarily be available. One approach that could overcome this problem is emergent language segmentation using Harris' Articulation Scheme, recently employed by Ueda et al. (2023). Ueda et al. (2023) compute the conditional entropy of each character in the emergent language, segmenting the messages where the conditional entropy increases. However,

even after language segmentation, there is no easy way to interpret the segments, as no method has been proposed to map them to specific meanings.

We present an approach to both segment the emergent language and map the segments to their meanings. We use a collocation measure called Normalised Pointwise Mutual Information (NPMI) (Bouma, 2009), often used in computational linguistics (Lim and Lauw, 2024; Thielmann et al., 2024; Yamaki et al., 2023). It is used to determine which messages are used for which observations and to analyse how the messages are composed, including whether they are trivially compositional (Korbak et al., 2020; Perkins, 2021; Steinert-Threlkeld, 2020). By applying a collocation measure to different parts of each message as well as the whole message, we can address the problems of both segmentation and interpretation of the message segments. This approach allows any part of the message to carry a different meaning. For example, if an emergent message contains segments that frequently appear in contexts involving specific integers, NPMI can help identify these segments and their meanings based on their statistical association with those integers.

NPMI is a normalised version of the Pointwise Mutual Information (PMI) (Church and Hanks, 1989), which is a measure of association between two events. PMI is widely used in computational linguistics, to measure the association between words (Han et al., 2013; Paperno and Baroni, 2016). Normalising the PMI measure results in its codomain being defined between $-1$ and $1$, with $-1$ indicating a purely negative association (*i.e.*, events **never** occurring together), $0$ indicating no association (*i.e.*, events being **independent**), and $1$ indicating a purely positive association (*i.e.*, events **always** occurring together). Normalised PMI is used for convenience when defining a threshold at which we consider a message or $n$-gram to carry a specific meaning, as the threshold can be between $0$ and $1$, instead of unbounded numbers in the case of PMI. [3]

To determine which parts of each message are used for a given meaning, two algorithms are proposed.

1. $PMI_{nc}$ The algorithm to measure non-compositional monolithic messages, most often used for target positional information (*e.g.*, *begin+1* (Section 2)); and

2. $PMI_c$ the algorithm to measure trivially compositional messages and their $n$-grams, used to refer to different integers in different positions.

A visual representation of the different types of messages that the algorithms can identify is provided in Figure 2. The $PMI_{nc}$ algorithm can identify any non-compositional messages, while the $PMI_c$ algorithm identifies both position variant and invariant compositional messages. The positional variance of the emergent language means that the position of an $n$-gram in the message also carries a part of its meaning. In this work, $n$-grams refer to a contiguous sequence of n integers from the sender's message. Consequently, in one message there are 3 unigrams ($m_1$, $m_2$, $m_3$), two bigrams ($[m_1, m_2]$, $[m_2, m_3]$), and one trigram (*i.e.*, the whole message $[m_1, m_2, m_3]$).

Figure 2 shows that in the position invariant case, the bigram $[5, 6]$ always carries the meaning of 4. While in the position variant case, the bigram $[5, 6]$ in position 1 of the message means 4, but $[5, 6]$ in position 2 of the message means 8. This can also be interpreted as the position of the bigram containing additional information, meaning a single "word" could be represented as a tuple of the bigram and its position in the message, as both contribute to its underlying information. Non-compositional messages are monolithic, *i.e.*, the whole message carries the entire meaning. For example, message $[5, 6, 8]$ means the target is in the first position, while $[5, 6, 6]$ means the target is one to the right of 9, even though the two messages share the bigram $[5, 6]$.

**The $PMI_{nc}$ algorithm**   The $PMI_{nc}$ algorithm calculates the NPMI per message by first building a dictionary of all counts of each message being sent, together with an observation that may provide positional information (*e.g.*, *begin+1*) or refer to an integer in a given position (*e.g.*, 1 left of the target). The counts of that message and the counts of the observation, including the integer position, are also collected. For example, consider the observation $\boldsymbol{o} = [4, -1, 15, 16, 13]$. For the corresponding message $\boldsymbol{m}$, the counts for each integer in each position relative to the target would increase by 1 (*i.e.*, $left1[4]+ = 1$, $right1[15]+ = 1$ *etc.*). The count for the message signifying *begin+1* would also be increased. Given these counts, the algorithm then estimates the probabilities of all respective events (messages, positional observations, and integers in given positions) and calculates the NPMI measure.

---

[3]Our implementation of NPMI is not numerically stable due to probability approximation, sometimes exceeding the [-1,1] co-domain. We provide more details in the code.

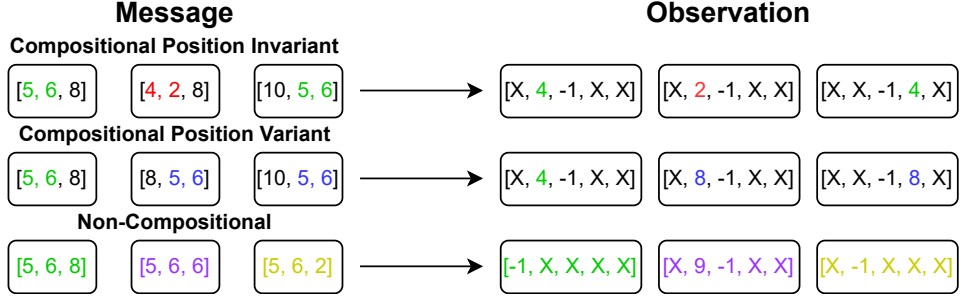

Figure 2: Examples of the different types of message compositionality that are possible to identify using the PMI algorithms.

**The PMI$_c$ algorithm**   The PMI$_c$ algorithm first creates a dictionary of all possible $n$-grams, given the message space ($m$) and maximum message length (3). The list of all possible $n$-grams is pruned to contain only the $n$-grams present in the agents' language, avoiding unnecessary computation in the later parts of the algorithm. Given the pruned list of $n$-grams, the algorithm checks the context in which the $n$-grams have been used. The occurrence of each $n$-gram is counted, together with the $n$-gram position in the messages and the context in which it has been sent, or the integers in the observation. The $n$-gram position in the message is considered to account for the possible position variance of the compositional messages.

Consider the previous example, with $o = [4, -1, 15, 16, 13]$ and a message $m = [11, 13, 5]$. For all $n$-grams ($[11], [13], [5], [11, 13]$, *etc*.) of the message, all integers are counted, irrespective of their positions (*i.e.*, $counts[4]+ = 1$, $counts[15]+ = 1$, *etc*.).

Given these counts, the PMI$_c$ algorithm estimates the NPMI measure for all $n$-grams and all integers in the observations. These probabilities are estimated from the dataset using the count of their respective occurrences divided by the number of all observations/messages.

Once the NPMI measure is obtained for the $n$-gram-integer pairs, the algorithm calculates the NPMI measure for $n$-grams and referent positions or the positions of the integer in the observation the message refers to. For example, given an observation $o = [4, -1, 15, 16, 13]$, if the message contains an $n$-gram which has been identified as referring to the integer $15$, the rest of the message (*i.e.*, the unigram or bigram, depending on the length of the integer $n$-gram) is counted as a possible reference to that position, in this case, to position $right1$, or 1 to the right of the target. This procedure follows for all messages, building a count for each time an $n$-gram was used together with a possible $n$-gram for an integer. These counts are used to calculate the NPMI measure for $n$-gram and position pairs.

The PMI$_c$ algorithm also accounts for the possible position invariance of the $n$-grams, *i.e.*, where in the message the $n$-gram appears. This is achieved by calculating the respective probabilities *regardless* of the position of the $n$-gram in the message, by summing the individual counts for each $n$-gram position.

**Pseudocode**   We provide a condensed pseudocode for both algorithms in Algorithm 1. In the case of the PMI$_{nc}$, the $n$-grams in the pseudocode would be whole messages, *i.e.*, trigrams. This base pseudocode would then be duplicated, interpreting the context as either an observation that may provide positional information (*e.g.*, *begin+1*) or an integer.

For the PMI$_c$ algorithm, only the unigrams and bigrams would be evaluated. The base pseudocode would also be duplicated, once for the integer in a given position, and second for the referent position. Each would be used as the context in which to evaluate the NPMI for each $n$-gram. A detailed commented pseudocode for both the PMI$_{nc}$ and PMI$_c$ algorithms is available in Algorithm 2 and Algorithm 3 in Appendix D, respectively.

Both algorithms use two hyperparameters: a confidence threshold $t_c$ and top_n $t_n$. The confidence threshold refers to the value of the NPMI measure at which a message or $n$-gram can be considered to refer to the given part of the observation unambiguously. To account for polysemy (where one symbol can have multiple meanings), the agents can use a single $n$-gram to refer to multiple integers.

---

**Algorithm 1:** PMI Algorithm Base

---

1 Gather `ngram_counts`, `context_counts`, `joint_counts`, `n_grams`;

2 **for** *each n-gram g in position p and context c* **do**

3      $P(g, p) = \texttt{ngram\_counts}[g] \cdot \frac{1}{\texttt{total } n\texttt{-grams}}$;

4      $P(c) = \texttt{context\_counts}[c] \cdot \frac{1}{\texttt{total contexts}}$;

5      $P(g, p; c) = \texttt{joint\_counts}[(g, c)] \cdot \frac{1}{\texttt{total } n\texttt{-grams}}$;

6      $\text{NPMI}(g, p; c) = \log_2 \frac{P(g, p, c)}{P(g)P(c)} \cdot \frac{1}{-\log_2 P(g, p, c)}$;

7 **end**

8 return NPMI;

---

This is given by the second hyperparameter, top_n, which sets the degree of the polysemy, or the number of integers to be considered for a given $n$-gram.

## 5 Spatial referencing experiments

The agent pairs are trained over 16 different seeds to verify the results' significance. All agent pairs achieve above $98\%$ accuracy on the referential task, showing that the agents develop a way to communicate about spatial relationships in their observations. The analysis provided in this section is based on the messages collected from the test dataset after the training has finished.

The two hyperparameters, $t_c$ and $t_n$ (Section 4), governing the NPMI measure have been determined through a grid search to maximise the understanding of the emergent language, by maximising the translation accuracy. The results in this section are obtained using the best-performing values for each of the hyperparameters. We provide the values for the grid search in Appendix A.

### 5.1 Emergence of non-compositional spatial references

Using the $\text{PMI}_{nc}$ algorithm, we detect the emergence of messages tailored to convey the positional information contained in the observations. As mentioned in Section 2, sender observations which require shifting convey additional information about the position of the target within the sequence. In over $90\%$ of agent pairs, these observations are assigned unique messages, used only for each kind of observation, *i.e.*, *begin*, *begin+1*, *end-1* and, *end*.

In $20\%$ of runs which develop these specialised messages, the same repeating character is used to convey the message. The characters used for these observations are *reserved* only for these kinds of observations. For example, in one of the runs the agents use character 11 to signify the beginning of the sequence, with the character 11 being used only in two contexts: as the messages $[11, 11, 11]$ to signify *begin*, or as a message $[0, 11, 11]$ to signify *begin+1*. In other cases, characters are fully reserved for specific messages. *e.g.*, 22 is used only for *end*, in the message $[22, 22, 22]$.

The emergence of non-compositional references used for other observations is also detected using the $PMI_{nc}$ algorithm. Such messages refer to a specific integer in a specific position of the sender observation, *e.g.*, $o_5 = 10$. While we allow for polysemy of the message in our analysis using $t\_n = [1, 2, 3, 5, 10, 15]$, we observe the highest translation accuracy with $t\_n = 1$, indicating that the non-compositional messages do not have any additional meanings.

### 5.2 Emergence of compositional spatial references

Using the $PMI_c$ algorithm, we also detect the emergence of *compositional spatial references* for 25% of agent pairs. Such messages are composed of two parts, a positional reference and an integer reference. The positional reference specifies where a given integer can be found in the observation, in relation to the masked target integer $-1$. The integer reference specifies which integer the positional reference is referring to. For example, one pair of agents has assigned the unigram 7 to mean that the *target* integer is 2 to the right of the *given* integer, and the bigram $[0, 2]$ to mean the integer 18.

Table 1: Average emergence and vocabulary coverage of all message types.

| Message Type | Avg. % Emergence | Avg. % of Messages |
|---|---|---|
| Non-Compositional Positional | 99.3% (100%-93.75%) | 1% (3%-0%) |
| Non-Compositional Positional Reserved | 18.75% (18.75%-18.75%) | 1% (3%-0%) |
| Non-Compositional Integer | 45.1% (100%-0%) | 10% (15%-0%) |
| Compositional Integer | 100% (100%-100%) | 34% (99.7%-0%) |
| Compositional Positional | 25% (27%-0%) | 56% (100%-0%) |

Together, a message can be composed $[7, 0, 2]$, which means that the target integer for the receiver to identify is 2 to the right of the integer 18, *i.e.*, $\boldsymbol{o} = [18, X, -1, X, X]$. This allows the sender to identify the target integer exactly, given the sequence $\boldsymbol{s}$.

In Table 1, we summarise the emergence of each type of message across all runs, together with the percentage of the vocabulary that they represent. The entries in the table are composed of average percentages, across all $t_n$ and $t_c$ choices. In the parentheses, we show the maximum and minimum values across all $t_n$ and $t_c$ choices. The average % of emergence represents the absolute % of runs which developed that message type or message feature. For all messages, the average % of messages which are of a given type or exhibit a given feature is only counted for in runs where these features emerged.

## 5.3 Evaluating interpretation validity and accuracy

To ensure the validity of our message analysis, we present two hypotheses which, if supported by the results, would indicate that the mappings generated by the NPMI measure are correct.

**Hypothesis 1 (H1)** If the correlations exist and do not require non-trivial compositionality (Perkins, 2021), and are not highly context-dependent (Nikolaus, 2023), then the evaluation accuracy should be significantly higher than chance, or above $20\%$, when using the identified mappings.

**Hypothesis 2 (H2)** If the positional components of compositional messages are correctly identified and carry the intended meaning, then their inclusion should result in an increase in accuracy.

Given the messages identified by the NPMI method, we test **H1** and **H2** by using a dictionary of all messages successfully identified, given a value of both NPMI hyperparameters $t_n$ and $t_c$. A dataset is generated to contain only targets which can be described with the messages present in the dictionary.

For the non-compositional messages, the dataset is generated by selecting a message from the dictionary at random, and creating an observation that can be described with that message. Given a non-compositional message that corresponds to the target being on the right of the integer 15, an observation $\boldsymbol{o} = [1, 15, -1, 5, 36]$ would be created. Analogously, for non-compositional positional messages such as *begin* an observation $\boldsymbol{o} = [-1, 15, 8, 5, 36]$ would be created.

For the compositional messages, we create the observations by randomly selecting a positional component and an integer component from the dictionary. For example, given the unigram 7 meaning that X is 2 to the left of the target, we could select the bigram $[8, 14]$ corresponding to the integer 30. The observation created could then be $\boldsymbol{o} = [30, 8, -1, 36, 5]$. The dataset creation process for the compositional messages also checks if the observations can be described given the two $n$-grams in their required positions within the message.

To test **H2**, a dataset is created using **only** the integers that can be described by the dictionaries, randomly selecting integer components from the dictionary, and creating the respective observations. This process also accounts for the required positions of the message components so that a message describing the observation can always be created. For example, if the unigram 9 described the integer 11, and the bigram $[5, 1]$ described the integer 6, a corresponding observation could be $\boldsymbol{o} = [11, 6, -1, 8, 9]$. The positions of the integers in the observations are chosen at random. By generating both compositional datasets using a stochastic process, we do not assume a specific syntax. Rather, the syntax can only be identified by looking at messages understood by the receiver.

Table 2: Accuracy improvements using the NPMI-based dictionary, $\pm$ denotes the 1-sigma standard deviation. Non-Compositional Positional refers to messages such as *begin* or *end*, Non-Compositional Integer refers to the non-compositional monolithic messages describing both the position and the integer, Compositional-NP refers to messages only containing the identified integer components, and the Compositional-P which refers to messages containing both the identified integer and positional components.

| Dict Type | $t_n$ | $t_c$ | Average Accuracy | Maximum Accuracy |
|---|---|---|---|---|
| Non-Compositional Positional | 1 | 0.9 | 90% $\pm$3% | **94%** |
| Non-Compositional Integer | 1 | 0.5 | 36% $\pm$0.4% | 37% |
| Compositional-NP | 1 | 0.5 | 22% $\pm$ 2% | 28% |
| Compositional-P | 1 | $0.5^3$ | 30% $\pm$ 21% | 78% |

These datasets, together with their respective dictionaries, are then used to query the receiver agent, testing if the messages are identified correctly. We run this test for all of our trained agents, with the dictionaries that were identified for each agent pair. We provide the details in Table 2.

Using just the non-compositional positional messages, we observe a significant increase in the performance of the agents, compared to random chance accuracy of **20%**. This proves **H1**, showing that at least some messages do not require complex functions to be composed, or contextual information to be interpreted. As the accuracy for these messages reaches over 90% on average, we argue that the NPMI method has captured almost all the information transmitted using these messages.

As mentioned in **H2**, we examine the impact of the positional components and whether they carry the information the NPMI method has identified. We, therefore, separate the compositional analysis into two parts: Compositional-NP, where the positional components are replaced with 0, and Compositional-P, which includes the identified positional components. In the Compositional-NP case, the agents achieve a close to random accuracy, whereas, in the Compositional-P case, agents achieve above random accuracy, with some agent pairs reaching over 75% accuracy. This proves our **H2** correct, showing that the NPMI method has successfully identified the positional information contained in the messages, together with the integer information.

## 6   Discussion

Having successfully verified both **H1** and **H2**, we confirmed the validity of the language analysis. We also verify the generalisation ability of the agents, by evaluating varying training and evaluation sequence lengths, vocabulary sizes, and hidden size in Appendix C.

To provide human interpretability of the emergent language, we use the NPMI method to create a dictionary providing an understanding of both the positional and compositional messages. We present an excerpt from an example dictionary in Table 3. With human interpretability, we can gain a deeper understanding of the principles underlying the agents' communication protocol.

We posit that the emergence of compositional spatial references points to a first emergence of a simple syntactic structure in an emergent language. Both of the $n$-grams in our example from Section 5.2, also shown in Table 3, are assigned specific positions in the message by the agents. The unigram 7 must always be in the first position of the message, while the bigram $[0, 2]$ must always be in the second position. The emergence of this structure shows that even though referential games have been considered obsolete in recent research (Chaabouni et al., 2022; Rita et al., 2024), a careful design of the environment may yet elicit more of the fundamental properties of natural language.

We hypothesise that the emergence of non-compositional spatial references tailored to specific observations, such as *begin+1*, is due to observation sparsity. Compositionality would bring no benefit since the observations which they describe are usually rare, representing 1-2% of the dataset and are monolithic, *i.e.*, *begin*, *begin+1*, *end-1*, and *end*. We therefore argue that the emergence of non-compositional references in these cases is **advantageous**, since these messages are easily compressible. Since these messages are monolithic, they could be compressed to a single token/character in

---

[3] $t_n$ for the referent position $n$-grams is set to 0.3

simple encoding schemes. In contrast, compositional messages require at least two tokens/characters, one for each integer/positional component. With a linguistic parsimony pressure (Chaabouni et al., 2019; Rita et al., 2020) applied, these messages could be more efficient at transmitting the information contained within these observations than compositional ones.

Table 3: Example dictionary of the agents' messages and their meanings

| Message | Type | Meaning |
|---|---|---|
| $[11, 11, 11]$ | Non-Compositional Positional | *begin* |
| $[0, 11, 11]$ | Non-Compositional Positional | *begin+1* |
| $[10, 10, 10]$ | Non-Compositional Positional | *end-1* |
| $[18, 18, 18]$ | Non-Compositional Positional | *end* |
| $[12, 16, 14]$ | Non-Compositional Integer | 15 is 1 left of target |
| $[15, m_2, m_3]$ | Compositional Positional | ? is 2 left of target |
| $[7, m_2, m_3]$ | Compositional Positional | ? is 2 right of target |
| $[m_1, 0, 17]$ | Compositional Integer | Integer 1 |
| $[m_1, 0, 2]$ | Compositional Integer | Integer 18 |
| $[m_1, 8, 14]$ | Compositional Integer | Integer 30 |

## 7 Limitations

The accuracy for the Non-Compositional Integer, and Compositional-P messages averages about 33%. While still above random, showing that some meaning is captured in non-compositional messages, it points to there being more to be understood about these messages. We hypothesise this may be due to the higher degree of message pragmatism, or context dependence (Nikolaus, 2023). Our method of message generation, using randomly selected parts, may not be able to capture the complexity of the messages. For example, the context in which they are used might be crucial for some $n$-grams, requiring the use of a specific n-gram instead of another when referring to certain integers, or when specific integers are present in the observation. Just like in English, certain verbs are only used with certain nouns, such as "pilot a plane" vs "pilot a car". While the word "pilot" in the broad sense refers to operating a vehicle, it is not used with cars specifically. This may also be the case for the emergent language. For compositional messages, an additional issue may be that some messages are non-trivially compositional, using functions apart from simple concatenation to convey compositional meaning (Perkins, 2021), making them impossible to analyse with the NPMI measure. However, these issues may be addressed by scaling the emergent communication experiments as the languages become more general with the increased complexity of their environment (Chaabouni et al., 2022).

## 8 Conclusion

Recent work in the field of emergent communication has advocated for better alignment of emergent languages with natural language (Boldt and Mortensen, 2024; Rita et al., 2024), such as through the investigation of deixis (Rita et al., 2024). Aligned to this approach, we provide a first reported emergent language containing *spatial references* (Lyons, 1977), together with a method to interpret the agents' messages in natural language. We show that agents can learn to communicate about spatial relationships with over 90% accuracy. We identify both compositional and non-compositional spatial referencing, showing that the agents use a mixture of both. We hypothesise why the agents choose non-compositional representations of observation types which are sparse in the dataset, arguing that this behaviour can be used to increase communicative efficiency. We show that, using the NPMI language analysis method, we can create a human interpretable dictionary, of the agents' own language. We confirm that our method of language interpretation is accurate, achieving over 94% accuracy for certain dictionaries.

## Acknowledgments and Disclosure of Funding

This work was supported by the UK Research and Innovation Centre for Doctoral Training in Machine Intelligence for Nano-electronic Devices and Systems [EP/S024298/1].

The authors would like to thank Lloyd's Register Foundation for their support.

The authors acknowledge the use of the IRIDIS High-Performance Computing Facility, and associated support services at the University of Southampton, in the completion of this work.

For the purpose of open access, the authors have applied a CC-BY public copyright licence to any Author Accepted Manuscript version arising from this submission.

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

## A    Training Details

The computational resources needed to reproduce this work are shown in Table 4, with the hyperparameters in Table 5 and Table 6. The Table 4 shows resources required for all training and evaluation. The processors used were a mixture of Intel Xeon Silver 4216s and AMD EPYC 7502s. The GPUs used were a mixture of NVIDIA Quadro RTX 8000s, NVIDIA Tesla V100s, and NVIDIA A100s. These nodes used in our experiments were hosted on the IRIDIS cluster. The development process consumed more compute, which we estimate would have added 10 CPU and GPU hours, to account for experimentation.

Table 4: Compute resources

| Resource | Value (1 Run) | Value (Training Total) | Value (Evaluation & Analysis) |
|---|---|---|---|
| Nodes | 1 | 8 | 1 |
| CPU | 16 cores | 128 cores | 64 cores |
| GPU | 1 | 8 | 1 |
| Memory | 50 GB | 400 GB | 120 GB |
| Storage | 1 GB | 32 GB | 32 GB |
| Wall time | 2 hours | 240 hours | 24 hours |

Table 5: Hyperparameters

| Parameter | Value |
|---|---|
| Epochs | 1000 |
| Optimizer | Adam |
| Learning Rate $\alpha$ | 0.001 |
| Gumbel-Softmax Temperature | [1.0] |
| Training Dataset Size | 200k |
| Test Dataset Size | 20k |
| No. Distractors | 4 |
| No. Points | [20,40,60,100] |
| Message Length | 3 |
| Vocabulary Size | [13,26,52] |
| Sender Hidden Size | [64,128] |
| Receiver Hidden Size | [64,128] |

Table 6: PMI Grid Search Parameters

| Parameter | Values |
|---|---|
| $t_c$ | [0.1, 0.2, 0.3, 0.4, 0.5, 0.6, 0.7, 0.8, 0.9] |
| $t_n$ | [1, 2, 3, 5, 10, 15] |

## B    Dataset Details

To train and evaluate the agents, we use datasets consisting of 200,000 samples for training, 200,000 for validation, and 20,000 for testing. Each dataset is generated independently, with sequences created randomly. Given the sequence length of 60 and the fact that no integers are repeated, the number of possible permutations is $60! \approx 8 \times 10^{81}$, which vastly exceeds the number of samples we generate. We further ensure that there is no overlap between datasets by empirically checking the overlap rates across 1,000 randomly generated datasets, confirming an overlap rate of 0%.

## C    Generalisation

To generalise the results presented in this paper, we also run additional tests, varying the vocabulary size, training sequence length, evaluation sequence length, and the hidden size of the agents, as

outlined in Appendix A. We observe no performance decline with either increasing or decreasing the vocabulary size or the training sequence length, given that the agents have enough capacity within their network to still learn the longer sequence lengths. We observe a decline in task accuracy at sequence lengths of 100, when the agents have a hidden size of 64. However, increasing the hidden size to 128 brings the training and validation accuracy back to over 90%.

When agents are evaluated on sequence lengths that are different from the ones they were trained on, we observe a small performance decline for small differences in sequence lengths. We present the average accuracies for the base case (Sequence shortened by 0), as well as the average difference in accuracy as compared to the baseline for different sequence lengths in Table 7. We observe a significant difference if the agents are evaluated on sequences that are over 50% shorter than the ones they were trained on. We hypothesise that this is due to the agents missing certain integers that they used more often than others, therefore reducing their accuracy. However, even in the worst case, the accuracy remains above 70%.

Table 7: Evaluation of different sequence lengths

| Training sequence length | Sequence shortened by | | | | |
|---|---|---|---|---|---|
| | 0 | -5 | -10 | -20 | -40 |
| 20 | 98.68% | -0.53% | -7.50% | N/A | N/A |
| 40 | 95.59% | 0.05% | -0.75% | -5.55% | N/A |
| 60 | 92.98% | 0.30% | -0.30% | -2.47% | -15.8% |
| 100 | 86.23% | 0.34% | -0.03% | -1.26% | -5.2% |

## D  Algorithm Descriptions

For our pseudocode we will be using the Python assignments convention, *i.e.*, $=$ and $\leftarrow$ are equivalent, and $x$+=1 is equivalent to $x \leftarrow x + 1$. The algorithms presented are for $top\_n = 1$. To improve the computational efficiency. the probability of the integer appearing is statically defined as $\frac{1}{60}$ for $top\_n = 1$, or in Equation (1) for $top\_n > 1$. In the case of $top\_n > 1$ we use the probability for the integer as per Equation (1), to account for the polysemy, *i.e.*, the probability for any of $top\_n$ integers occurring in the observation. The lower part of the binomial is 4, as there are 4 integers that can be sampled from the 60 possible integers, instead of 5, as we exclude the target integer.

$$p(integers) = \frac{\binom{60}{4} - \binom{60-top\_n}{4}}{\binom{60}{4}} \qquad (1)$$

Additionally, in the PMI$_c$ algorithm, we specify a probability to equal to 0.98 in Line 74 and Line 77. This is a simplification of the calculation for clarity of the pseudocode. This probability is instead obtained using the count of a given type of observation, divided by the number of total observations. This calculation is performed for each type of observation, *i.e.*, *begin*, *begin+1*, *end*, *end-1* and *middle*. The probability of the *middle* observation is very close to 1, being on average 0.98, while the other probabilities are on average 0.005. Since the *middle* observation is most common, we included its value in the pseudocode.

**Algorithm 2:** The PMI$_{nc}$ algorithm

---

**Data:** $O\_M$ ;                                      # All observations together with sent messages
**Data:** $L = len(O\_M)$ ;                          # Total number of observations with sent messages
**Data:** $S = [begin, begin + 1, end - 1, end]$ ;            # List of positional observations
**Result:** $pmi_{nc}[m][NPMI]$

1   $pmi_{nc} = \text{dict}$;
2   **for** $o, m \in O\_M$ **do**
3      $pmi_{nc}[m][count]$ += 1 ;                          # Message occurrences
4      **for** $pos \in S$ **do**
5         **if** $o == pos$ **then**
6            $pmi_{nc}[pos][count]$ += 1 ;                      # Positional observations count
7            $pmi_{nc}[m][pos]$ += 1 ;                        # Message sent with positional observation
8         **end**
9      **end**
10     **for** $integer \in o$ **do**
11        $pmi_{nc}[m][integer\_pos][integer]$ += 1 ;            # Message sent with integer in given position
12     **end**
13 **end**
14 **for** $pos \in S$ **do**
15     $posit_{total} = pmi_{nc}[pos][count]$ ;                  # Count of positional observations
16     $p(pos) = \frac{posit_{total}}{L}$;                        # Estimate observation probability
17     **for** $m \in pmi_{nc}[m]$ **do**
18        $m_{total} = pmi_{nc}[m][count]$ ;                      # Total count of message
19        $ms_{total} = pmi_{nc}[m][pos]$ ;                    # Total count of message with positional obs
20        $p(m) = \frac{m_{total}}{L}$ ;                        # Estimate message probability
21        $p(m, pos) = \frac{ms_{total}}{L}$ ;                  # Estimate joint probability
22        $h(m, pos) = -\log_2(p(m, pos))$ ;
23        $pmi(m, pos) = \log_2(\frac{p(m,pos)}{p(m)p(pos)})$;
24        $npmi(m, pos) = \frac{pmi(m,pos)}{h(m,pos)}$;
25        $pmi_{nc}[m][NPMI] = npmi(m, pos)$;
26     **end**
27 **end**
28 **for** $pos \in pmi_{nc}[m]$ **do**
29     **for** $integer \in pmi_{nc}[m][pos]$ **do**
30        $p(pos) = \frac{1}{60}$;                      # Estimated observation probability for 60 integers
31        $m_{total} = pmi_{nc}[m][count]$ ;                      # Total count of message
32        $ms_{total} = pmi_{nc}[m][pos][integer]$ ;   # Total count of message with integer in given position
33        $p(m) = \frac{m_{total}}{L}$ ;                        # Estimate message probability
34        $p(m, pos) = \frac{ms_{total}}{L}$ ;                  # Estimate joint probability
35        $h(m, pos) = -\log_2(p(m, pos))$ ;
36        $pmi(m, pos) = \log_2(\frac{p(m,pos)}{p(m)p(pos)})$;
37        $npmi(m, pos) = \frac{pmi(m,pos)}{h(m,pos)}$;
38        $pmi_{nc}[m][pos][integer][NPMI] = npmi(m, pos)$;
39     **end**
40 **end**

---

**Algorithm 3:** The $PMI_c$ algorithm

---

**Input:** $t_c$ ;                                                                      # Confidence value
**Data:** $O\_M$ ;                                          # All observations together with sent messages
**Data:** $L = len(O\_M)$ ;                                 # Total number of observations with sent messages
**Data:** $ngrams$ ;                                       # List of all message $n$-grams present in $O\_M$
**Result:** $pmi_c[m][NPMI]$

1  $pmi_c$ = dict;
   ; # First we identify $n$-grams corresponding to integers.
2  **for** $ngram \in ngrams$ **do**
3   **for** $o, m \in O\_M$ **do**
4    **if** $ngram \in m$ **then**
5     $pmi_c[ngram][count]$ += 1 ;                              # Total $n$-gram occurrences
6     $pmi_c[ngram][ngram\_pos][count]$ += 1 ;      # $n$-gram occurrences including $n$-gram
         position
7     **for** $integer \in o$ **do**
8      $pmi_c[ngram][integer][count]$ += 1 ;   # $n$-gram sent with integer in given position
9      $pmi_c[ngram][ngram\_pos][integer][count]$ += 1 ; # $n$-gram in given position sent
          with integer in given position
10    **end**
11   **end**
12  **end**
13 **end**
   ; # Calculate integer NPMI.
14 **for** $ngram \in ngrams$ **do**
    ; # Position variant NPMI.
15  **for** $pos \in pmi_c[ngram][ngram\_pos]$ **do**
16   $p(integer) = \frac{1}{60}$;                        # Estimated observation probability for 60 integers
17   $integer_p = max(pmi_c[ngram][integer][count])$;;                      # Find integer with highest
        co-ocurrence given position
18   $ngram_{pos} = pmi_c[ngram][ngram\_pos][count]$ ;
19   $p(ngram_{pos}) = \frac{ngram_{pos}}{L}$
20   $p(ngram_{pos}, integer) = \frac{pmi_c[ngram][ngram\_pos][integer][count]}{L}$;
21   $h(ngram_{pos}, integer) = -\log_2(p(ngram_{pos}, integer))$;
22   $pmi(ngram_{pos}, integer) = \log_2(\frac{p(ngram_{pos}, integer)}{p(ngram_{pos})p(integer)})$;
23   $npmi(ngram_{pos}, integer) = \frac{pmi(ngram_{pos}, integer)}{h(ngram_{pos}, integer)}$;
24   $pmi_c[ngram][ngram\_pos][integer] = npmi(ngram_{pos}, integer)$;
25  **end**
    ; # Position invariant NPMI.
26  $integer = max(pmi_c[ngram][integer][count])$;;      # Find integer with highest co-ocurrence
27  $p(integer) = \frac{1}{60}$;                         # Estimated observation probability for 60 integers
28  $ngram_{total} = pmi_c[ngram][count]$ ;
29  $p(ngram) = \frac{ngram_{total}}{L \times (4 - len(ngram))}$ ;     # If $n$-gram is length 1, it could appear 3 times per message
30  $p(ngram, integer) = \frac{pmi_c[ngram][integer][count]}{L}$;
31  $h(ngram, integer) = -\log_2(p(ngram, integer))$;
32  $pmi(ngram, integer) = \log_2(\frac{p(ngram, integer)}{p(ngram)p(integer)})$;
33  $npmi(ngram, integer) = \frac{pmi(ngram, integer)}{h(ngram, integer)}$;
34  $pmi_c[ngram][integer] = npmi(ngram, integer)$;
35 **end**

---

**Algorithm 4:** The PMI$_c$ algorithm cont.

; # Now we identify $n$-grams corresponding to referent positions.

36  $ngram_{pr}$ = dict;

   ; # Prune $n$-grams with NPMI below $c$

37  **for** $ngram \in pmi_c$ **do**

38     **for** $integer \in pmi_c[ngram]$ **do**

39        **if** $pmi_c[ngram][integer] < t_c$ **then**

40           del $pmi_c[ngram][integer]$;

41        **end**

42        **for** $pos \in pmi_c[ngram]$ **do**

43           **for** $integer \in pmi_c[ngram][pos]$ **do**

44              **if** $pmi_c[ngram][pos][integer] < t_c$ **then**

45                 del $pmi_c[ngram][pos][integer]$;

46              **end**

47           **end**

48        **end**

49     **end**

50  **end**

   ; # Find messages with integer $n$-grams

51  **for** $ngram \in pmi_c[ngram]$ **do**

52     **for** $o, m \in O\_M$ **do**

      ; # Position variant $n$-gram

53        **if** $pmi_c[ngram][pos]$ **then**

54           **if** $ngram \in m[pos]$ **then**

55              $new\_ngram = m - ngram$;              # Get leftover $n$-gram

56              $pr = pos(pmi_c[ngram][pos][integer], msg)$ ;  # Get the possible referent position

57              $ngram_{pr}[new\_ngram][pr][count] += 1$ ;    # Count leftover $n$-gram occurence

58              $ngram_{pr}[new\_ngram][pos][pr][count] += 1$ ; # Count leftover $n$-gram occurence
            in given positions

59           **end**

60        **end**

      ; # Position invariant $n$-gram

61        **else**

62           **if** $ngram \in m$ **then**

63              $new\_ngram = m - ngram$ ;             # Get leftover $n$-gram

64              $pr = pos(pmi_c[ngram][integer], msg)$ ;    # Get the possible referent position

65              $ngram_{pr}[new\_ngram][pr][count] += 1$ ;    # Count leftover $n$-gram occurence

66              $ngram_{pr}[new\_ngram][pos][pr][count] += 1$ ; # Count leftover $n$-gram occurence
            in given positions

67           **end**

68        **end**

69     **end**

70  **end**

**Algorithm 5:** The PMI$_c$ algorithm cont.

; # Calculate referent position NPMI.

71 **for** $ngram \in ngram_{pr}$ **do**

72    **for** $pr \in ngram_{pr}[ngram][pr]$ **do**

      ; # Position variant NPMI.

73       **for** $pos \in ngram_{pr}[ngram][pos][pr]$ **do**

74          $p(pr) = 0.98$;           # Estimated observation probability for given position

75          $ngram_{pos} = ngram_{pr}[ngram][pos][pr][count]$ ; $p(ngram_{pos}) = \frac{ngram_{pos}}{L}$

         $p(ngram_{pos}, pr) = \frac{ngram_{pr}[ngram][pos][pr][count]}{L}$ ;

         $h(ngram_{pos}, pr) = -\log_2(p(ngram_{pos}, integer))$;

         $pmi(ngram_{pos}, pr) = \log_2(\frac{p(ngram_{pos}, pr)}{p(ngram_{pos})p(pr)})$;

         $npmi(ngram_{pos}, pr) = \frac{pmi(ngram_{pos}, pr)}{h(ngram_{pos}, pr)}$ ;

         $pmi_c[ngram][pos][pr] = npmi(ngram_{pos}, pr)$;

76       **end**

      ; # Position invariant NPMI.

77       $p(pr) = 0.98$;           # Estimated observation probability for given position

78       $ngram = max(ngram_{pr}[ngram][pr][count])$ ;   # Find highest positional reference count

79       $p(ngram) = \frac{ngram}{L}$;

80       $p(ngram, pr) = \frac{ngram_{pr}[ngram][pr][count]}{L}$;

81       $h(ngram, pr) = -\log_2(p(ngram, integer))$;

82       $pmi(ngram, pr) = \log_2(\frac{p(ngram, pr)}{p(ngram)p(pr)})$;

83       $npmi(ngram, pr) = \frac{pmi(ngram, pr)}{h(ngram, pr)}$;

84       $pmi_c[ngram][pr] = npmi(ngram, pr)$;

85    **end**

86 **end**

