# OpenReview forum: "Speaking Your Language: Spatial Relationships in Interpretable Emergent Communication"
_NeurIPS.cc/2024/Conference — NeurIPS 2024 poster_

### Official Review · Reviewer_vZ26 · 2024-07-01

**Soundness:** 4
**Presentation:** 3
**Contribution:** 2
**Rating:** 6
**Confidence:** 4

**Summary:**

This work investigates the presence of spatial deixis (e.g., spatial references in language dependent on the context of the utterance) in a signalling game within the paradigm of emergent communication.
It begins by introducing a variant of the signalling game which requires the sender to communicate the relative position of some element in a sequence of integers visible to the receiver.
Analysis of the semantics of the emergent communication is performed primarily through looking at normalized point-wise mutual information.
These analyses show that the emergent communication regularly uses spatially-referent messages (or sub-message units), validating the presence of spatial deixis in the environment.

**Strengths:**

In conjunction with standard criteria, there are three characteristics that are particularly important for emergent communication research: reusability (how easily can another researcher use the products of this research), generalizability (how much do the findings of this research apply broadly to our knowledge of emergent communication), and directedness (does this research contribute concretely to particular questions in emergent communication research).

### Quality
- (major) The experimental design is of good quality; the methods are in line with the standards of the field.
### Clarity
- (major) The language of this paper is easy to read and illustrates the central points effectively.
### Reusability
- (major) The provided code appears to be of good quality (have not attempted to run it); this work would be very easy to reuse in subsequent experiments.
### Generalizability
- (minor) The environment is relatively simple, with few confounding factors, making it easier to draw conclusions about more general tendencies in EC.
### Directedness
- (major) This paper is directed toward an important goal in emergent communication: discovering/engineering more human-like features in EC, namely deixis or context-dependent reference.
- (minor) Secondarily, this paper also demonstrates some degree of compositional semantics and syntactic features.

**Weaknesses:**

### Quality
- Nothing of note.
### Clarity
- (minor) One or two more of the points in Section 5 need to be illustrated (likely with just a table); although the text is mostly clear, some tables which aggregate what is said there would make referencing the paper much easier.
- (minor) It is a little bit confusing that your hypotheses are what you expect to be false; it might be clearer to state hypotheses in positive (even if the statistical test you are using is rejecting a null hypothesis of a random baseline).
### Reusability
- Nothing of note.
### Generalizability
- (major) It is not clearly the case that the environment addresses deixis in a way that applies to emergent communication environments more generally (see first question for more).
- (major) There is not much discussion on how the deixis investigated in this paper is applicable to emergent communication more generally.
### Directedness
- Nothing of note.

**Questions:**

- The biggest question regarding the paper for me is how do we make the jump for the simple example of deixis presented in the empirical investigation of the paper to a more robust form of deixis.  It is not wrong (and likely correct, in fact) to have started out with a toy problem, but I believe there either needs to be empirical work and/or some light theoretical work on what exactly is meant by "deixis" in this paper, how the environment investigated satisfies that definition, and how this is relevant for further environments.  I might raise my score if the authors could respond with a slightly more formal characterization of deixis and how it maps both to the current environment and more sophisticated, "natural" environments (e.g., embodied multi-agent environment).
- In the environment, are the integers actually represented as integers in the neural network or are they encoded as one-hot vectors?  If they are OHVs, it is not clear that it is the case; if they are not OHVs, it seems like an odd design choice to feed a scalar into an NN when it is representing something categorical.
- How are the various inputs to the receiver agent actually fed into the network.  Are they just concatenated "temporally" and given as a sequence to the GRU?


### Comments

- The fact that this paper is inducing a segmentation of emergent communication is (minor) contribution in and of itself, so I think it deserves a mention in the introduction.
- The "td" variable needs to be introduced before the example; I am assuming it is the list of distractors plus the correct answer, but it should be stated explicitly when defining the vectors earlier.
- Table 2: Don't reuse X; use other variables.
- Include the actual URL for the (Anonymous) GitHub so that it is obvious it is a link.
- I don't understand the point being made at Line 337.
- I do not think it is appropriate to specifically mention "SVO" as an interpretation of the language since there is no clear way to distinguish between nouns, verbs, subjects, or objects; I think it is fine to say that there is syntactic structure, but I a skeptical of there being evidence to make any claim further than that here.

**Limitations:**

N/A.

---

> ### Author Rebuttal · Authors · 2024-08-06
>
> # Response to Reviewer vZ26
>
> We thank the reviewer for the insightful feedback and constructive criticisms. We appreciate that the reviewer found our experimental design and overall approach to be of high quality and relevant to the field of EC. We also appreciate that the code was found to be of high quality.
>
> To address the comments of the reviewer:
>
> ## Questions
>
> > The biggest question regarding the paper for me is how do we make the jump for the simple example of deixis presented in the empirical investigation of the paper to a more robust form of deixis. (...)
>
> Let $O$ be the observation available to an agent. Let $O_p$ be a point in the observation from which the relative position will be described. For example, $O_p$ can represent the agent position or the position of an object. Let $O_t$ be the target object to which the relative position is being described, for example, a different object or agent. Let $d(O_t)$ denote the relative distance of $O_t$ to the reference point $O_p$. $\theta(O_t)$ is defined as the angle between $O_p$ and $O_t$, using for example the vector angle.
>
> The spatial deixis could then be defined as a mapping of tuples $(d, \theta)$ to specific $n$-grams. For example, a tuple $(2,\pi)$ can be mapped to the $n$-gram $15$ from this paper, representing "2 to the left". This, combined with specifying the $O_t$, completes the message, which in this paper could be $[15,0,2]$ to mean "$O_p$ is to 2 to the left of 15".
>
> The version of deixis presented in this paper is a special case of this general deixis, where $O$ is a 1D tensor, $O_p$ is prescribed to be always $-1$, and the $\theta$ can only take values of $\pi$ and $0$, as we operate in 1D. However, the concepts presented can be easily extended to the case of 2D or 3D tensors and observations, and when the agent chooses both $O_p$ and $O_t$. When extended to multi-agent settings, the agents may need to additionally specify their position, or both $O_p$ and $O_t$ in their messages, since their observations may have different relative positions.
>
> > In the environment, are the integers actually represented as integers in the neural network or are they encoded as one-hot vectors? (...)
>
> The integers in the environment are indeed represented as scalars. Based on your comment, we have run additional tests of the OVH approach and found that it performed significantly worse. Instead of learning to communicate about their observations, the agents appeared to be memorising the dataset, increasing accuracy on the training set but staying at random on the validation dataset. Instead, in the scalar approach, the agents learn to communicate, increasing their accuracy on the validation set. This test was performed on agents with the same hidden sizes, except for the layers processing the one-hot vectors. It may be the case that with more tuning of the layer sizes the OVH approach would work, but this is outside the scope of this paper. An additional advantage of using the scalar approach are savings in terms of the number of trainable parameters (98.8K vs 72K), with about a 27% reduction in size.
>
> > How are the various inputs to the receiver agent actually fed into the network. (...)
>
> Let $B$ be the batch size, $T$ the temporal/sequence length dimension, $C$ the hidden size of the GRU and Linear layers (which are the same), $N_v$ the vocabulary size of the agents, and $N_l$ the message length.
>
> The output of the sender agent is in the form of $[B,N_l,N_v]$ , or $B$ tensors, each representing the probability of a character being in a given position of the message, as per the Gumbel-Softmax reparametrisation.
>
> This is fed into the receiver agent, to `GRU 1` in Figure 1, from which only the hidden states for the last character probabilities are collected, i.e., only the hidden states after each message has been processed in full. This results in a tensor of shape $[1,B,C]$, where $1$ represents the number of layers of the GRU.
> This tensor can then be used as the initial hidden state for `GRU 2` in Figure 1.
>
> `GRU 2` receives the sequences in the form of $[B,T,1]$, with $T$ being our sequence length, and the $1$ representing the scalar values. We again collect only the final hidden states for each element, resulting in the output of shape $[1,B,C]$.
>
> The $td$ vector is passed through a linear layer, in shape $[B,5,1]$, with $5$ representing the 4 distractors and 1 correct target. Similarly to the `GRU 2` input, we use scalar values. The linear layer outputs a tensor of shape $[B,5,C]$.
>
> Finally, the `GRU 2` output is matrix multiplied with the $td$ embeddings. We first permute the `GRU 2` output tensor, creating a tensor of shape $[B,C,1]$.
> Then the two tensors are multiplied, i.e., $[B,5,C] \times [B,C,1]$, resulting in a tensor of shape $[B,5,1]$, which is then squeezed to produce the batch of target predictions $[B,5]$.
>
> ## Comments
>
> > I don't understand the point being made at Line 337:
>
> We apologise for the confusion. We argue that non-compositional messages are easily compressible, when they would be transmitted in a real-world setting, with limited bandwidth. For simple encodings, since these messages are monolithic, they could be compressed to a single token/character. In contrast, compositional messages require at least two tokens/characters, one for each integer/positional component. We will clarify this in the revised version.
>
> > I do not think it is appropriate to specifically mention "SVO" as an interpretation of the language since there is no clear way to distinguish between nouns, verbs, subjects, or objects. (...)
>
> We have removed the claim of SVO ordering from the paper. Instead, we discuss a possible syntactic structure, but make no comparisons to natural language syntax or concepts, as reviewer `VwFJ` suggested.

---

> > ### Comment · Reviewer_vZ26 · 2024-08-09
> > **Reply**
> >
> > Regarding the response to the "generalization" point of the review starting with "Let O be the observation available to an agent", I think this is a step in the right direction but still needs more development (although it is a little hard to judge the proposed formalization out of context).
> > I think the main shortcoming of the sketch of formalization proposed is that it is still not quite clear what "deixis" is in a sort of environment-agnostic way, which is what I was getting at in my review (although potentially unclearly).
> >
> > For example, $O_p$ is a point in the observation serving as the origin, but what does "point in the observation" mean in some environment-independent way?
> > Maybe one could start off by specifying that the environment has some structure which supports deictic expressions in the first place (e.g., space, time, succession, 1st-2nd-3rd person distinction).
> > Additionally, there needs to be some way that the context in which the potentially deictic expression is grounded; that is, we can't just have some arbitrary $O_p$, there needs to be some way in which $O_p$ is derivable from the context of the utterance.
> > With these in mind, we could maybe define deixis in an EC environment along the lines of the following:
> > - The observations of the environment support some notion of position (or something more general, if you can think of it).
> > - Using this notion of position, we can take some original point, apply a transformation (i.e., moving a distance in space), and derive the target point.
> > - The original point is derived from the context of the utterance.
> >
> > I am not saying this the best or only way to do, but hopefully this better illustrates what I meant by "generalization".  I think the account offered in the rebuttal is on the right track, but maybe a couple extra steps could be taken to make it environment agnostic.
> > I am moving recommendation from a 5 to a 6 because think the proposed changes are close to making this paper more widely applicable to other EC environments.
> > Nevertheless, I probably will move my rating to a 7 because I am not able to see a fully-developed formalization in context.
> >
> > ---
> >
> > As an aside, regarding OHVs vs scalars, it seems odd that OHVs would perform worse since it essentially pre-processes the scalar values into something the neural networks can work with---maybe this has something to do with over-parameterization when switching from scalars to OHVs, but I don't think this all that important (aside from making it clear that scalars are being used since one might naturally assume integers are represented as OHVs), as it does not invalidate the experiments.
> >
> > Everything else not mentioned here I found was addressed suitably by the rebuttal.

---

> > > ### Author Response · Authors · 2024-08-12
> > >
> > > Thank you for your continued engagement with our paper! We improve the formalisation of deixis below, making it more environment-agnostic.
> > >
> > > To do so, we expand the observation to be an $m$-dimensional $R^n$ tensor, which can represent any possible properties of the observation.
> > >
> > > ## Formalisation of deixis
> > >
> > > Let $O$ represent an abstract observation that an agent perceives from its environment, $O \in R^{n \times n \times n ... n = n^m}$. The $m$ dimensions can represent the spatial, temporal, or other positions. For example $R^{n^3}$ could represent a 3-D observatrion.
> > >
> > > Let $O_p$ be the reference point in $O$, which coordinates are represented by an $n$-tuple of real numbers $(x_1,x_2 ... x_m)$ and $O_t$ be the target point in $O$, with its coordinates represented by an $n$-tuple of real numbers $(y_1, y_2 ... y_m)$.
> > >
> > > Then, the relative distance function $d(O_p,O_t)$ returns an $n$-tuple of real numbers $(z_1,z_2 … z_m)$, such that $z_i = x_i - y_i$. This relative distance function allows for unambiguous identification of the target object $O_t$, given that the position of $O_p$ is known.
> > >
> > > A deictic expression can be defined as a mapping of the value of $d(O_p,O_t)$, the reference point $O_p$, and their context $O$, to a specific linguistic or symbolic expression that describes the relationship between $O_p$ and $O_t$. This mapping can be represented as:
> > >
> > > $(O,d(O_p,O_t),O_p) \rightarrow \text{Expression}(O,d(O_p,O_t),O_p)$
> > >
> > > where the resulting expression $\text{Expression}(O,d(O_p,O_t),O_p)$ is a description of the reference point $O_p$ and its relative distances to the target point $O_t$, given the context $O$.
> > >
> > > ### Examples
> > >
> > > Consider two objects, $O_p$ and $O_t$, located at Cartesian coordinates $(x_1, x_2)$ and $(y_1, y_2)$, respectively. The relative distance function $d(O_p, O_t) = (z_1, z_2)$ captures the differences in both dimensions.
> > >
> > > In this context, suppose the expression "2 to the left and 3 up" is used. This would correspond to $z_1 = -2$ (indicating 2 to the left) and $z_2 = 3$ (indicating 3 up) relative to the reference point $O_p$. However, without knowing the exact position of $O_p$, this information alone is insufficient to identify $O_t$.
> > >
> > > To resolve this, the reference point $O_p$ must either be explicitly included in the deictic expression or implicitly understood by all interlocutors. For example, if it is agreed that the reference point is at $(5, 4)$, where $(5,4)$ could be an abstraction of any object, the expression "2 to the left and 3 up" clearly locates $O_t$ at $(3, 7)$. Without such an agreement, a full deictic expression is necessary, such as "2 to the left and 3 up from $(5, 4)$".
> > >
> > > Depending on the context $O$ and the need for clarity between agents, some information provided by the relative distance function $d$ may be omitted. For example, if the target object $O_t$ is the only item of a given type in the environment to the left of the reference point $O_p$, simply stating "to the left" might be sufficient without specifying the exact distances or the $y$ direction. Similarly, if the agents are conversing about objects on a flat surface, the height difference may be irrelevant and ignored, simplifying the expression further.
> > >
> > > The examples above could also be rewritten to showcase the ability to specify any other deictic basis, by simply changing the underlying meaning of a given dimension. The expression "5 minutes after 10:00" specifies a temporal deixis by explicitly locating an event $O_t$ relative to a reference time $O_p$. In a 3D spatial context, an expression like "4 to the left, 2 forward, and 1 down" captures the exact relative position of an object $O\_t$ in Euclidean space.
> > >
> > > This formalization will be included in the revised version of our paper.

---

> > > > ### Comment · Reviewer_vZ26 · 2024-08-13
> > > >
> > > > Thank you for providing this.  It is pretty close to what I had in mind in writing my prior comment, and I think it will facilitate applying the results of this paper more broadly.

---

### Official Review · Reviewer_JyuC · 2024-07-12

**Soundness:** 3
**Presentation:** 3
**Contribution:** 3
**Rating:** 7
**Confidence:** 4

**Summary:**

This paper proposes a new communication game in the emergent communication framework to analyze the emergence of _deictic reference, i.e. expressions akin to demonstratives like "this" and "that".  These are important expressions in natural language and especially in this emergence literature, since their meaning is context-dependent and "functional", i.e. cannot be reduced to objective properties of the object of reference. The paper also introduces an application of normalized pointwise information to the analysis of the emergent communication protocol in order to identify holistic messages (where a message refers to an entire meaning) and compositional ones (where certain n-grams and/or positions refer to specific "components" of the meaning).  Both of these are welcome contributions and will be of interest to many people working on emergent communication.  The core idea in their game is to use integers within longer sequences as the object of reference, provide a _partial observation_ of the true context to the sender (so that absolute positional information cannot be used) and to _mask out_ the target object in a sequence (so that the integer itself cannot be used); what remains as possible information to convey are things like "two to the right of 13".

**Strengths:**

* A carefully designed emergent communication scenario which requires something like spatial deixis to emerge for successful communication.  This is an important component of human language that goes beyond what has been done in existing literature.
* Interesting and useful application of NPMI for the analysis of (non-)compositionality of the resulting messages.
* Engages well with existing literature to situate the new contribution of this paper.
* Results also show a robustness to things like random seed, which is not always the case in emergent communication.

**Weaknesses:**

* Some experimental details could be more carefully reported and some analyses could be more systematic/quantitative (see questions below).
* The artificial messages used to validate their NPMI metric does not yield results as strong as one would like (as discussed in the Limitations section); this makes it not entirely clear that the metric does what its intended to do.

**Questions:**

* Why did you choose a fixed-length of 3 for the messages (as opposed to either a single token, or variable-length)?
* Line 92 and 114: should "target integers" and "targets" both be singular?  There's one target integer, correct?  Or is the plural here just over a batch of examples?  If the latter, the wording is a bit confusing since the worked case in the paper is just one example ("batch size 1" so to speak).
* Can $PMI_c$ and $PMI_{nc}$ be seen as one metric, with the latter a special case of the former (i.e. for the full tri-grams)?  The discussion just before Section 5 seems to suggest so, so I would encourage more elaboration on whether these are really two separate degrees or not.  For instance: does high $nc$ entail low $c$, and vice versa?
* "The analysis provided in this section is based on the messages collected from the test dataset after the training has finished".  What was the train/test split here?  Appendix A provides model / optimizer hyper-parameters, but what are the game/environment/data choices?
* Can the observations in Section 5.1 and 5.2 be made more quantitative?  I would appreciate a more detailed analysis of the types of composition observed, their frequency, and other factors like that.
* H2 and Table 1: while the Comp-P case is above chance, if the NPMI method correctly identified "genuinely compositional" messages, we would expect nearly perfect accuracy in this case, right?
* Very minor typographic point: I think that the "n" in "$n$-gram" and similarly in the main text should be in math mode.

**Limitations:**

Yes

---

> ### Author Rebuttal · Authors · 2024-08-06
>
> # Response to Reviewer JyuC
>
> We thank the reviewer for their comprehensive comments and thoughtful critique. We appreciate that the reviewer has found our paper interesting and that the investigation of deictic references is perceived to be of high value.
>
> To address the points raised by the reviewer:
>
> ## Weaknesses
>
> > Some experimental details could be more carefully reported and some analyses could be more systematic/quantitative (see questions below).
>
> > The artificial messages used to validate their NPMI metric does not yield results as strong as one would like (as discussed in the Limitations section); this makes it not entirely clear that the metric does what its intended to do.
>
> Both answered below.
>
> ## Questions
>
> > Why did you choose a fixed-length of 3 for the messages (as opposed to either a single token, or variable-length)?
>
> The choice was motivated by our goal of analysing the spatial deixis and the analysis method. A single token would not be able to be segmented and could not show any possible compositionality. While it would be possible for the agents to use a single token to convey the same information, assuming a large vocabulary size, such tokens could not be interpreted as easily. Additionally, allowing for variable length messages increases the computation cost of the analysis. As our method checks all possible $n$-grams used by the agents, increasing the message length would correspondingly increase the number of $n$-grams needing to be generated. The special case handling of each message would also add overhead. However, extending our setting to variable-length messaging, both the training and analysis code, is entirely possible and feasible.
>
> > Line 92 and 114: should "target integers" and "targets" both be singular? There's one target integer, correct? Or is the plural here just over a batch of examples? If the latter, the wording is a bit confusing since the worked case in the paper is just one example ("batch size 1" so to speak).
>
> Thank you for pointing this out. There is indeed only one target integer. The camera-ready version will correct this wording.
>
> > Can $PMI_c$ and $PMI_{nc}$ be seen as one metric, with the latter a special case of the former (i.e. for the full tri-grams)? The discussion just before Section 5 seems to suggest so, so I would encourage more elaboration on whether these are really two separate degrees or not. For instance: does high $nc$ entail low $c$ , and vice versa?
>
> $PMI_c$ and $PMI_{nc}$ can be viewed as complementary metrics. High $nc$ values for certain messages do not necessarily entail low $c$. While in practice the two metrics do align in opposition, there could be cases where if a compositional message is most often used only in a specific context, it could have both high $nc$ and $c$ values. For example, a message $[1,9,10]$, where $1$ is a positional component, and $[9,10]$ are integer components, could always be used whenever the integer represented by $[9,10]$ is in a given position. Then, even though the agents use compositional rules to create this message, it would also be classed as high $nc$, since its parts are never observed separately. It would also have a high $c$, as the aforementioned $n$-grams are also associated with the given integer and position.
>
> As mentioned, in practice, compositional parts of the compositional messages will be frequently used in different messages, lowering the value for $nc$.
>
> > "The analysis provided in this section is based on the messages collected from the test dataset after the training has finished". What was the train/test split here? Appendix A provides model / optimizer hyper-parameters, but what are the game/environment/data choices?
>
> The datasets used are: training 200k, validation 200k, test 20k. We generate each dataset separately instead of splitting a singular dataset into the 3 used. We are confident that this approach does not present any issues with overlap, as all sequences are randomly generated. Considering the sequence length of 60, since there are no integer repetitions, the number of permutations is $60! \approx 8 \times 10^{81}$, far outnumbering our number of generated samples. We also empirically confirm this by checking the overlap across 1000 randomly generated datasets, finding an overlap rate of 0%.
>
> We will provide additional information about the environmental and dataset choices in the appendices of the revised version.
>
> > Can the observations in Section 5.1 and 5.2 be made more quantitative? (...)
>
> We assume this is similar to the comment by Reviewer `vZ26`. We provide this additional quantitative information in table format in the general response. It will also be included in the camera-ready version.
>
> > H2 and Table 1: while the Comp-P case is above chance, if the NPMI method correctly identified "genuinely compositional" messages, we would expect nearly perfect accuracy in this case, right?
>
> If the compositional messages were context-free and composed only by concatenation of the positional components, we would indeed expect nearly perfect accuracy.
>
> The lower accuracy for the Comp-P case might be due to contextual information not accounted for by our creation of the messages or the possibility that some messages may require slightly more complex composing methods. For example, if a certain integer component representing $5$ is only used when $6$ is also present in the observation, our method of creating the messages would fail. Accounting for such context dependence is not encoded in the message segmentation or generation process. Similarly, if there are "synonyms" for meanings such as "1 to the left" that are only used in certain contexts, this would also not be accounted for in our methods.

---

> > ### Comment · Reviewer_JyuC · 2024-08-12
> >
> > Thank you for the detailed and helpful reply!  I am already a fan of this paper, and these responses clarify any last remaining questions that I had.

---

### Official Review · Reviewer_VwFJ · 2024-07-15

**Soundness:** 4
**Presentation:** 3
**Contribution:** 2
**Rating:** 5
**Confidence:** 4

**Summary:**

The authors design a referential game environment intended to motivate the emergence of spatial references, cast in the form of a task where the target integer must be selected from an integer sequence.  The character vocabulary for the message is smaller in size than the set of integers in the list, and this necessitates an alternative to directly specifying the target integers.  Using a traditional GRU-based speaker/listener architecture, the model achieves high task accuracy.  Using existing information theoretic measures the authors are able to roughly decode the semantics of the messages and show some degree of compositionality in the messages.

**Strengths:**

- The experimental design is simple but I think straightforward and correct for what the authors want to test.

- Similarly, it appears from the analysis in sections such as Table 2 that the resulting messages do seem to exhibit a variety of communication strategies, including the desired type in some messages (compositional positional)

- The approach of decoding the meaning and segmentation of the messages via NPMI (though I would have also liked to see some discussion of where this would/would not be appropriate in terms of a general evaluation metric for EC.  It seems some strong expectation over what the emergent language needs to say may be necessary?  In this case, the presence of the integers, for instance)

**Weaknesses:**

- Overall I find the biggest weakness to be in the scope of the paper and the degree to which the design of the environment caters to the type of messages the authors want to elicit here.  It comes across as a toy problem, and through the lens of the field as a whole, I think it raises the question of whether there is sufficient novelty in making such small and targetted tweaks to the referential game formula.

This might be best highlighted by revisiting the motivating example, such as "a blue vase with intricate motifs on the table".  Why refer to this as "the vase over there" or "that vase near you"?  There are pressures in the referential game to draw out these spatial references, but these feel artificial and devoid of broader understanding about linguistic pressures when we compare them to the pragmatic concerns that would motivate a spatial reference (and what type of spatial reference) in the motivating example.

I also think claims are over-stated.  The authors claim this is the first paper in EC to have syntax and make comparisons to SVO ordering.  This comes across as a flimsy attempt to connect to human language and to signal a degree of progress in the complexity of ECs, but a lengthier discussion is warranted.  Syntax can't be treated as something that does or does not exist, but rather, discussion of what formal language class the emergent language falls into would be relevant, and nothing here would necessitate a CFG or the degree of syntax that is meaningful when it comes to discussing natural language.

Despite the simplicity I do see value in this work but I would have liked to have seen a less contrived environment with a more difficult learning problem / substantial scope in the necessary semantics to feel comparable to the degree of contributions typical of a paper at this venue.  I think it would be far more appropriate at a more targetted venue where it can also recieve appropriate attention and discussion.

**Questions:**

- There are some fairly trivial solutions to this problem.  It seems compositional integer style gets at this -- are there cases where compositional integer would fail?  I'm not seeing the need to learn a spatial solution to this problem when it seems that a two-character code could cover all possible target integers.  Does this vary as the length of sequence or size of alphabet are increased/decreased?

- Similarly, is there any reason to motivate the choice of vocabulary size with respect to latin alphabet?  The chunks of the messages are more akin to words than characters.  To me it just read as an attempt to have a connection to human language, but that relationship was not meaningful.

- It is mentioned that the hyperparameters for MI are optimized for translation accuracy.  I can make some guesses as to what might be done here, but it wasn't clear to me exactly what is being compared here.

**Limitations:**

Some limitations have been discussed, though limitations of the experimental design (such as 3 character messages, "nouns" being integers and therefore being able to reference without features, etc.) are not.  The authors mention some doubts in previous work regarding the value of referential game -- I think this is precisely one of the ways that these doubts emerge, i.e., "we try very hard to setup a task where success must be achieved in this way, and the model finds it".  These sorts of limitations may not be addressable in the scope of the paper, but they are also not mentioned.  There are no negative societal impacts from such work.

---

> ### Author Rebuttal · Authors · 2024-08-06
>
> # Response to Reviewer VwFJ
> We would like to thank the reviewer for the insightful feedback and detailed criticism. We also appreciate that the reviewer found our experimental setup and analysis interesting.
>
> To address the points raised by the reviewer:
>
> ## Weaknesses
> > Overall I find the biggest weakness to be in the scope of the paper and the degree to which the design of the environment caters to the type of messages the authors want to elicit here. It comes across as a toy problem, and through the lens of the field as a whole, I think it raises the question of whether there is sufficient novelty in making such small and targetted tweaks to the referential game formula. (...)
>
> We address this weakness in detail in the general response.
>
> > I also think claims are over-stated. The authors claim this is the first paper in EC to have syntax and make comparisons to SVO ordering. This comes across as a flimsy attempt to connect to human language and to signal a degree of progress in the complexity of ECs, but a lengthier discussion is warranted. (...)
>
> We agree that the claim of SVO ordering may be an overstatement, so we have removed it from the paper. Instead, we point to a possible syntactic structure, but make no comparisons to natural language syntax, as also suggested by reviewer `vZ26`.
>
> ## Questions
> > There are some fairly trivial solutions to this problem. It seems compositional integer style gets at this -- are there cases where compositional integer would fail? I'm not seeing the need to learn a spatial solution to this problem when it seems that a two-character code could cover all possible target integers. Does this vary as the length of sequence or size of alphabet are increased/decreased?
>
> We apologise for the confusion — the compositional integers style does include spatial referencing, where one character represents an integer, and the other the relative position of that integer. It represents a spatial solution. Simply representing the target integer is impossible, as the sender does not know the target integer, so that cannot be transmitted. If the sender, instead of using the compositional message, always transmits the identity of the integer one to the left of the target, we would still consider this a spatial solution. The spatial deixis is implicitly agreed between the sender and the receiver. Indeed, for our setting, a two character code, which includes spatial referencing, would be enough to convey all information. However, without explicit or implicit spatial referencing, we do not identify a feasible solution to this environment.
>
> We do not observe any differences in terms of the performance of our agents when the train sequence length is increased/decreased, except for the convergence speed and need to adjust the hidden layer sizes, as noted in our response to Reviewer `YdNG`. Similarly, there is no observable difference for changes in the alphabet sizes, up to a point: there must be enough characters, and the message space must be large enough for the sender to be able to describe its observations. In the case of making the message space too small, the sender cannot convey enough information about its observations to the receiver. If the message space is too large, the sender can just encode the complete observation, bypassing the need for communication.
>
> We will include this information in the revised version of the paper.
>
> >  Similarly, is there any reason to motivate the choice of vocabulary size with respect to latin alphabet? The chunks of the messages are more akin to words than characters. To me it just read as an attempt to have a connection to human language, but that relationship was not meaningful.
>
> The choice of vocabulary size was arbitrary, with $26$ providing high expressivity and presenting a good starting point. If such a vocabulary is enough to convey information in natural languages like English, this should also apply to the agents. We have additionally run preliminary tests with smaller and larger sizes of the vocabulary and found no impact on the agent performance.
>
> > It is mentioned that the hyperparameters for MI are optimized for translation accuracy. I can make some guesses as to what might be done here, but it wasn't clear to me exactly what is being compared here.
>
> We apologise for the lack of clarity. The optimised hyperparameters are $t_c$ and $t_n$, on which a grid search is performed. We create all possible permutations of the tuple ($t_c$, $t_n$), for $t_c \in [0.1, 0.2, 0.3, 0.4, 0.5, 0.6, 0.7, 0.8, 0.9]$ and $t_n \in [1, 2, 3, 5, 10, 15]$. Then, we select the best tuple of hyperparameters regarding the measured translation accuracy on the test dataset. We will clarify this in the camera-ready version.

---

> > ### Comment · Reviewer_VwFJ · 2024-08-13
> >
> > Regarding the simple environment, I agree and disagree.  Of course there is prudence in starting from a simple, well-controlled environment, as presented in the paper.  But of course the risk is that it will be difficult to scale the approach up or reconcile it with existing work.  Frankly this is an all too common problem in the EC literature, which is rampant with one-off papers which each show some semblance of the target structures in emergent languages, and which are never incorporated or built upon ever again, and cited only in passing as yet another linguistic phenomena which now has a toy world which can give rise to some simple form of it.
> >
> > The question should probably be, is it too much to ask for both?  Is it too much to expect that an EC paper at least show some consideration for what comes next, or to attempt to show both a contrived toy scenario and a more sophisticated one which explores the limitations of the approach in the face of a more realistic and more difficult learning problem?
> >
> > To this end, I greatly appreciate the prodding by Reviewer vZ26, and the author response, to formalize a more general notion of deixis, such that we could at least have some understanding of what scaling this work up might entail.
> >
> > With that in mind, with the inclusion of this formalisation I am willing to increase my score to 5, on the accept side of the decision boundary, as I think the work as a whole might represent an adequately thorough investigation of this specific issue.  This is a difficult decision -- I am still a bit concerned about the potential lack of impact from the work, especially when submitted to a competitive and general interest venue, and where the additions to the draft (which seek to formalize the broader phenomenon of deixis) are still a work in progress.

---

### Official Review · Reviewer_YdNg · 2024-07-17

**Soundness:** 3
**Presentation:** 2
**Contribution:** 2
**Rating:** 5
**Confidence:** 4

**Summary:**

This paper shows that emergent communication can learn spatial references. They first create a modified referential game which requires the agents to communicate by messages that indicate the relative position of a number. The proposed agent architecture shows that the GRU-based agents can achieve good performance. The analysis uses NPMI to identify the meaning of the ngrams in the message (i.e., the correlation between the ngram and the referred positions). This paper further shows that the mapping generated by NPMI is correct by generating additional datasets based on the identified dictionaries to show that both non-compositional and compositional messages carry the intended meanings.

**Strengths:**

- This paper proposes a novel spatial game to study the emergence of spatial references.
- This paper shows that NPMI is an effective measure to decompose the messages by finding correlations with the intended meanings.

**Weaknesses:**

- The paper is not very easy to follow especially the definition and design of different types of sequences, examples of the messages, and how the hypotheses are tested. The presentation can still be improved.
- It is unclear how much the test set overlaps with the training set when measuring the accuracy. There is no control of generalization tests such as varying full sequence length or observation of certain patterns of sequences. So, it is hard to understand if the learned messages are effective or memorization of part of sequences in training. For example, does the ngram that means “leftmost” can effectively communicate in a longer sequence?
- The design of the game put high communication pressure on the agents. The agents need to develop messages conveying relative positions in order to succeed. How does the success relate to the communication protocol, for example, when the message length is longer, is it still necessary to develop messages that convey relative positions? It is unclear about the role of channel bandwidth, effective communication, and developed messages.
- The test in Compositional-NP is to generate the dataset by removing the positional component of the message. This is an extreme case of H2. In reality, the message is most likely to be corrupted rather than removed. To reject the null hypothesis, it will be more convincing to have a corrupted message version.

**Questions:**

- In the experiment, the observation is always fixed length which makes the communication easier. What happens when the observation is longer? I can imagine if the longer sequence contains the same number at different positions, it will introduce some ambiguity in the messages.

**Limitations:**

- The experiment is based on one-dimensional sequences, the types of spatial references are limited in this case. It will be helpful if the author can discuss how it can extend to more complex types of spatial references.

---

> ### Author Rebuttal · Authors · 2024-08-06
>
> # Response to Reviewer YdNG
> We would like to thank the reviewer for their insightful comments and feedback. We appreciate that the reviewer found our game setting novel and the measure effective.
>
> We address the concerns and weaknesses raised below.
>
> ## Weaknesses
> > The paper is not very easy to follow especially the definition and design of different types of sequences, examples of the messages, and how the hypotheses are tested. (...)
>
> Based on your and the other reviewers' comments, we will improve the presentation for the camera-ready version of the paper. We will include the formalisations of the definitions, rephrasing the hypotheses, additional information about the dataset splits, and the types of different sequences tested.
>
> > It is unclear how much the test set overlaps with the training set when measuring the accuracy. There is no control of generalization tests such as varying full sequence length or observation of certain patterns of sequences. (...)
>
> The test set does not overlap with the train set or the validation set. Therefore, as our agents achieve high accuracy on the validation set, we can conclude that the results are not a case of overfitting. We will add the dataset split and overlap information in the revised version.
>
> We tested train sequence lengths between 20 and 100 and observed that agents can still learn to achieve the same accuracy as in the original task. The only differences are that longer sequences take longer to converge and require a larger hidden size.
>
> Longer sequences in the test dataset will be difficult to reliably evaluate after the training has finished due to the lack of trained weights for either the sender or the receiver to process given integers. This is a similar issue to extending the context windows in LLMs. However, sequences shorter than the ones present in the training dataset can be evaluated, and would contain integers that the agents would have the corresponding weights for. Our preliminary tests suggest that the agents can understand shorter sequences than the ones they were trained on. They can still achieve the same high accuracy as on the sequences from the training set. We only observe at most a 5-10% decrease in accuracy when the training and test sequences are significantly different lengths (60 vs 20 respectively). We hope this alleviates some concerns about the generalisation of our approach.
>
> Both tests will be expanded and included in the revised version of the paper.
>
> > The design of the game put high communication pressure on the agents. The agents need to develop messages conveying relative positions in order to succeed. How does the success relate to the communication protocol, for example, when the message length is longer, is it still necessary to develop messages that convey relative positions? (...)
>
> Channel bandwidth will indeed play a role in how the agents learn. Increasing the channel bandwidth to the point that the agents can describe the observation in full will nullify any development of spatial referencing.
>
> However, this is not the focus of this work. We show that the agents can communicate about spatial relationships, how such messages can be composed, and that this does not require as much bandwidth as transferring the complete observation. Therefore, analysing the bandwidth of the channel was outside the scope of this paper.
>
> >   The test in Compositional-NP is to generate the dataset by removing the positional component of the message. This is an extreme case of H2. (...)
>
> We apologise for the confusion. We think that our test already represents your suggestion. When the positional components are removed from the messages, any part of the message that does not convey information about an integer is replaced with $0$ instead of truncation. This ablation test is the most straightforward way of rejecting the null hypothesis of whether the positional components were correctly identified. We will clarify this in the camera-read version.
>
> ## Questions
> > In the experiment, the observation is always fixed length which makes the communication easier. What happens when the observation is longer? (...)
>
> As mentioned in our response to the weaknesses, we notice no differences when the observation length is increased.
>
> However, including integer repetitions would be challenging for the current setting and would require a careful design of the dataset. It would not provide additional information about the emergence of spatial deixis. If repetitions were allowed, there could be cases where the sender's observations could be duplicated multiple times in a single sequence. This would make the task of the receiver nearly impossible to accomplish, and would lead to training instability. Instead, to allow for repetitions, the sequence window would have to be extended, or the dataset generation would have to be designed to ensure that the sequence windows cannot be repeated while maintaining the possibility of single integer repetitions.
>
> To illustrate this, consider a sequence $S = [1,2,3,4,5,1,2,4,4,5]$, where each integer can be repeated multiple times. If the observation to the sender is $o_s=[1,2,-1,4,5]$, then it is impossible for the receiver to correctly predict the target number, as both $3$ and $4$ are valid answers.
>
> After applying such restrictions, the dataset allowing repetitions would most likely lead to the same results as the one presented in this paper. Possible differences include the sender having to refer to 2 integers in a single message to identify the referent position in the sequence precisely.
>
> ## Limitations
> > The experiment is based on one-dimensional sequences, the types of spatial references are limited in this case. It will be helpful if the author can discuss how it can extend to more complex types of spatial references.
>
> Thank you for this suggestion. We discuss the formal extension of our setting in our response to Reviewer `vZ26`.

---

> > ### Comment · Reviewer_YdNg · 2024-08-12
> > **Thanks for the reply**
> >
> > I thank the authors for the further explanation. It is good to see it generalize to a longer sequence. However, this is not the only form of generalization; also, non-overlapping training and test sets do not mean the messages generalize to different combinations of deixis. For example, seeing "4 to the left" and "3 to the right" and generalizing to "3 to the left". These generalization tests will need to reflect what a formal deixis definition this paper uses.
> >
> > Regarding the design to avoid repetition, I understand the difficulty in identifying the referent position, but this restriction can limit the learned messages to simple ones. So, it will be helpful to discuss this limitation in the design too.
> >
> > Overall, I appreciate the authors' response and it addressed some of my questions. I will increase my score to 5 and suggest a further discussion on the generalization that reflects the definition of deixis.

---

### Author Rebuttal · Authors · 2024-08-06

# General Response

We would like to thank the reviewers for their constructive comments.

We address some common themes in the general response, with more detailed comments in each reviewer rebuttal. Where it was needed, the quoted parts of the review texts were shortened to (...) for brevity. We welcome the reviewers' comments on any major unaddressed points in the discussion. We will be able to answer anything that was missing from our rebuttal.

We have also noted any typographical and layout comments, and assure the reviewers that they will be corrected in the revised version.

## Generalisation

Reviewers `YdnG` and `VwFJ` argue that the settings presented, and the results, are simple and not generalisable.

We agree that the environment present is quite simple. However, we disagree that this represents a weakness of our results. We see this as a good starting point, with any extraneous confounding factors removed, as reviewers `vZ26` and `JyuC` noted. By using a simple environment, we can show more precisely which factors affect the emergence of spatial referencing strategies. Using a more complex setting at such an early stage of research into deictic phrases in EC could potentially lead to confusing and not generalisable results. For example, using a vision network, or complex tasks, could lead to a generally low accuracy of the agents, which would not necessarily imply that spatial deixis are impossible, or require different approaches. Instead, it could point to an issue with the pretrained vision network, training the vision network, or the size of the agent's network being too small/large for the task given. We therefore consider that starting with the simple environment and showing how such spatial references can emerge, leads to useful insights, paving the way for future research in more complex settings with more complex architectures.

### Generalisation tests

We would also like to thank the reviewers for suggesting additional generalisation tests, allowing us to expand the analyses. We have run these preliminary tests, with the initial results showing that our methods generalise to training on both longer and shorter sequences. We also show that the agents can communicate even when presented with shorter sequences than the ones they were trained on. This indicates that the learned spatial references are transferrable to different environments, including OOD settings. These generalisation tests will be expanded on in terms of robustness and sample size, and featured in the camera-ready version of our paper.

## Presentation

All reviewers requested that the paper provide more information and formalisation, especially regarding the definitions and dataset details. We will include this in the camera-ready version. We will also be adding a more formal definition of deixis, as presented in our response to Reviewer `vZ26`.

Additionally, Reviewers `vZ26` and `JyuC` requested additional information for Section 5 of the paper. We present this in the table below. The entries in the table are composed of average percentages, across all $t_n$ and $t_c$ choices. In the parentheses, we show the maximum and minimum values across all $t_n$ and $t_c$ choices. The average \% of emergence represents the absolute \% of runs which developed that message type or message feature. For all messages, the average \% of messages which are of a given type, or exhibit a given feature, is only counted for in runs where these features emerged.

 This table will also be added to the camera-ready version.

| **Message Type**                                     | **Avg. \% Emergence**  | **Avg. \% of Messages** |
|-----------------------------------------------------|----------------------------------|---------------------------|
| Non-Compositional Positional      	       | 99.3\% (100\%-93.75\%) | 1\% (3\%-0\%)       	|
| Non-Compositional Positional Reserved | 18.75\%            	              | 1\% (3\%-0\%)       	|
| Non-Compositional Integer                       | 45.1\% (100\%-0\%) 	 | 10\% (15\%-0\%)     	|
| Compositional Integer             	        | 100\%              	            | 34\% (99.7\%-0\%)    	|
| Compositional Positional          	        | 25\% (27\%-0\%)    	| 56\% (100\%-0\%)     	|

---

### Decision · Program_Chairs · 2024-09-25

**Decision:**

Accept (poster)

**Comment:**

The paper proposes a variant of the referential game in which agents must learn to perform a simplified form of spatial deixis, by referring to a target item in terms of its relative position. The authors perform an information-theoretic analysis suggesting that the agents develop a language that is successfully incorporating deixis and is partially compositional.

The reviewers appreciated the interest and novelty of the research question. There were several concerns about generalization, for example in terms of length of the sequences, and a more general worry that the paper is only considering a toy setup for deixis, which might not be sufficient to build upon in future research.

There was a lively and constructive discussion, in which the authors addressed the length generalization issue, and they worked towards a formalization of deixis that establishes a clearer link between the paper setup and the general notion of deixis in linguistics.

The authors have addressed the main concern of the reviewers, and I think the revised paper will be a nice addition to the literature on emergent communication.